

# Tidal Variability of Chl-$a$ in the Indonesian Seas

Edward D. Zaron[1], Tonia A. Capuano[2], and Ariane Koch-Larrouy[2]

[1]College of Earth, Ocean and Atmospheric Science, Oregon State University, Corvallis, Oregon, USA
[2]LEGOS, France

**Correspondence:** Edward D. Zaron (edward.d.zaron@oregonstate.edu)

**Abstract.** Harmonic analysis of time series from 20 years of MODIS-Aqua ocean color observations (2002-2022) is conducted to identify periodic variability of near-surface chlorophyll (Chl-$a$) inferred from ocean color. As they are based on satellite imagery, the Chl-$a$ observations are characterized by significant gaps in both spatial and temporal coverage due to the masking of clouds in the images. Results yield a coherent picture of surface Chl-$a$ associated with the time mean, annual and semiannual cycles, and spring-neap tidal variability. Spring-neap variability is heterogeneous and associated with regions of significant baroclinic tides as well as coastal regions with strong tidal currents. The observations provide another line of evidence for the significant contribution of ocean tides to mixing in the Indonesian Seas.

## 1 Introduction

Turbulent transport caused by the disspation of internal tides is responsible for the transformation of Pacific Water into Indonesian Throughflow Water along transport pathways in the Indonesian Seas (Koch-Larrouy et al., 2007; Katavouta et al., 2022). The mixing is spatially heterogenous and is associated with sills and geographic features where baroclinic tides are particularly energetic in the main thermocline. Water mass properties, the associated sea surface temperature distribution, and resulting air-sea fluxes are strongly influenced by tidally-driven mixing in the Indonesian Archipelago, leading to broad impacts across the tropical climate system (Koch-Larrouy et al., 2010). Evidence for these processes has been obtained in a variety of studies using both in situ (Koch-Larrouy et al., 2015) and remotely-sensed data (Ray and Susanto, 2016).

The goal of this study is to document the variability of upper ocean chlorophyll (Chl-$a$) concentration associated with the spring-neap tides throughout the Indonesian Seas and nearby waters (from 23°S to 28°N, and 87°E to 148°E). Spring-neap variability arises when tidal currents associated with the sum of the $M_2$ and $S_2$ tides are rectified or otherwise interact to produce modulations of bottom stress or internal wave-driven mixing with a period of 14.77 days (denoted with the Darwin symbol, $MS_f$), phase-locked with the astronomical forcing. At least three different mechanisms could explain the association of remotely-sensed Chl-$a$ with spring-neap currents (Xing et al., 2021): a modulation of the tidally-driven turbulent flux of nutrients into the euphotic zone, a modulation of suspended sediment concentration affecting the intensity of photosynthetically-active radiation, and a modulation of resuspended Chl-$a$ associated with the microphytobenthos. The present study uses daily averages of Chl-$a$ inferred from multi-spectral imagery acquired by the MODIS-Aqua mission from 2002 to 2022 to describe Chl-$a$ variability. It is motivated by several ongoing studies which seek to model the physical and biological dynamics of the Indonesian Throughflow (Gutknecht et al., 2016; Nugroho et al., 2018).







**Figure 1.** The number of Chl-$a$ observations per pixel in the Level-3 GlobColour daily MODIS product. The rectangular regions indicate regions enlarged in subsequent figures, below. The maximum possible observation count, if there were no data gaps, is 7264 for the time period of the observations, July 2002 to August 2022.

Previous studies of Chl-$a$ variability employing satellite ocean color measurements have mentioned the challenges associated with data gaps due to cloudiness, sunglint, and water-type variability (Palacios, 2004; Susanto et al., 2006; Garcia and Garcia, 2008). A variety of approaches to overcoming these difficulties have been used to either fill data gaps (Yang et al., 2021) or phase-average the data according to days after full moon, for example, in the analysis of suspended sediment from ocean color (Shi et al., 2011). And, recently, another approach based on the wavelet analysis of 4.5 years of Himawari-8 imagery was used to study Chl-$a$ (Xing et al., 2021), which was capable of identifying seasonal and spring-neap variability. Given the



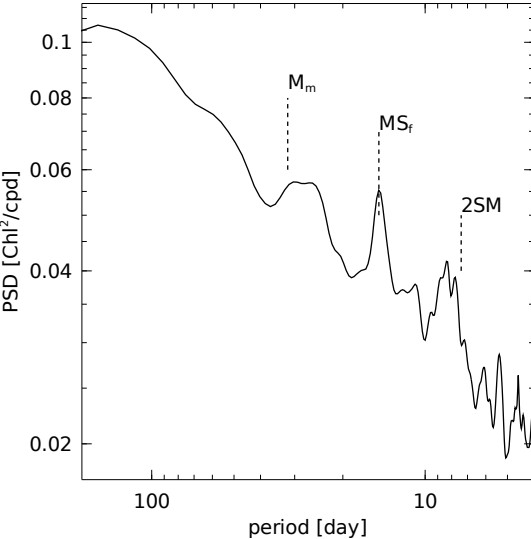

**Figure 2.** Spectral estimate of Chl-$a$ concentration off the northwest coast of Australia ($21°$S,$114°$E). The spectral estimate is computed by averaging Lomb-Scargle periograms based on Hamming-windowed data within 65-day segments. A narrow peak is evident at the MS$_f$ frequency (corresponding to 14.77 days, the M$_2$-S$_2$ sping-neap cycle), and broadband variance is present close to the first sub- and super-harmonics of MS$_f$ (denoted M$_m$ and 2MS, indicated). Harmonic analysis at the M$_m$ and 2MS frequencies, and at nearby compound overtide frequencies, did not find significant amplitude.

long, 20-year, duration of the MODIS record considered here, we have found that least-squares harmonic analysis is adequate to quantify variability, even with the data gaps and noise. A similar approach, which informed the development of the present

work, was successful in identifying spring-neap variability from 12 years of high-resolution sea surface temperature (SST) data in the Indonesian Seas (Ray and Susanto, 2016).

## 2 Data and Methods

The Chl-$a$ dataset used here consists of the Level-3 "GlobColour daily MODIS product," version 2018.4, which is an esti-
mate of the near-surface Chl-$a$ concentration in units of $\mathrm{mg/m^3}$ obtained from MODIS-Aqua data using a semi-analytical

model (the Garver-Siegel-Maritorena (GSM) product; Maritorena and Siegel 2005; Maritorena et al. 2010). The GSM product
is based on representing the spectrum of outgoing radiation with contributions due to Chl-$a$, dissolved organic matter, and
suspended particular matter, and estimating these quantities from a weigthed average of spectral data. The product provides
daily estimates; although, the constituent measurements may occassionally fall outside of the calendar date of the product. For
the purpose of harmonic analysis, the measurements are treated as if they occur at 12:00 GMT of product date. The spatial

resolution of the gridded GSM data is approximately 5 $\mathrm{km}$.

A map of the number of Chl-$a$ observations per pixel illustrates the influence of clouds and other processes responsible
for data gaps (Figure 1). The region just off the northwest coast of Australia contains the most data, the longest time series





comprising about 10 years of observations; although, this is only about 1/2 the total possible if every daily data value were populated. Individual pixel time series typically contain between 500 and 1000 data values.

The least-squares harmonic analysis of the Chl-$a$ is conducted using standard methods (Foreman et al., 2009). At each pixel a time series of Chl-$a$ is assembled, $\mathbf{y}$, and the design matrix, $\mathbf{A}$, is constructed with columns consisting of a constant (mean) and the cosine and sine of the astronomical arguments for annual ($S_a$, Doodson number 0 565 555), semi-annual ($S_{sa}$, Doodson number 0 575 555), and luni-solar fortnightly ($MS_f$, Doodson number 0 735 555) cycles. The vector of in-phase and quadrature harmonic constants, $\mathbf{x}$, is obtained as the least-squares solution of the linear system $\mathbf{y} = \mathbf{Ax}$.

Because of gaps in the time series, measurement noise, and the widely-varying magnitude of the tidal signals, error estimates are required to assess the significance of the mapped harmonic constants. Due to data gaps, the standard least-squares estimate computed from the spectrum of the residual is not applicable. Instead, the method of Matte et al. (2013) is used which is equivalent to the standard approach in the limit of gap-free data. The method uses the unitary discrete Fourier transform, denoted with matrix $\mathbf{F}$, to estimate a pseudo-spectrum of the residual. Given the residual, $\mathbf{r} = \mathbf{y} - \mathbf{Ax}$, a pseudo-spectrum

of $\mathbf{r}$ is defined as $S(\mathbf{r}) = \mathbf{Fr} \cdot (\mathbf{Fr})^*$, where $\cdot$ denotes the elementwise Shur product and super-script $^*$ denotes the complex conjugate. $S(\mathbf{r})$ is referred to as a pseudo-spectrum because it is computed from unevenly-spaced data; however, it is equal to the periodogram in the case of an evenly-spaced (gap-free) time series. After smoothing $S(\mathbf{r})$ to make the pseudo-spectrum continuous, the error variance of $\mathbf{x}$ is computed with a Monte-Carlo estimate of the matrix, $\mathbf{A}^{-1}\mathbf{F}^T diag(S(\mathbf{r}))\mathbf{FA}^{-1}$. This error estimate assumes that the covariance of the Fourier transform of the residual, $\mathbf{Fr}$, is diagonal, i.e., that the noise in $\mathbf{r}$ is

uncorrelated between pseudo-frequencies. This error estimate was compared to an approach mentioned in Ray and Susanto (2016), wherein the noise of the tidal harmonic constants was estimated by comparing it with the amplitude estimated at "fake" tidal frequencies, and found to agree well. In the plots shown below, the amplitude and phase of the Chl-$a$ harmonic constants are only plotted if the amplitude exceeds twice the estimated error level.

The choice of frequencies used in the harmonic analysis was informed by previous studies (Susanto et al., 2006; Ray

and Susanto, 2016; Xing et al., 2021) and trial-and-error. Figure 2 shows the power spectrum of Chl-$a$ from a region off the northwest coast of Australia. The spectral estimate was computed by averaging the Lomb-Scargle periodogram over ovelapping blocks of 60 observations (120 day time series, typically). The spectrum shows a narrowband peak at $MS_f$, and broadband variability close to the sub- and super-harmonics of this frequency (labelled $M_m$ and $2SM$, respectively), superimposed on a red spectrum. Harmonic analysis including the monthly and weekly harmonics, $M_m$ and $2SM$, did not show evidence of

phase-locked signals. Experiments with harmonic analysis of $\log_{10}(\text{Chl-}a)$ were conducted, but these did not reveal significant differences in the goodness-of-fit or significance compared to Chl-$a$, so the latter is shown below.

## 3   Results

Figure 3 shows the results of harmonic analysis for the mean Chl-$a$ concentration and the amplitudes of the $S_a$, $S_{sa}$, and $MS_f$ signals. Comparing the mean and $S_a$ (panels (a) and (b)), it is evident that the seasonal variation of Chl-$a$ is 50% or

more of the mean in many coastal areas, specifically, in the northeast Bay of Bengal, in the northeast Andaman Sea, in the





**Figure 3.** The (a) mean and amplitudes of (b) $S_a$, (c) $S_{sa}$, and (d) $MS_f$ from harmonic analysis of Chl-$a$. Two shades of gray are used to indicate land (dark gray) and non-significant amplitude compared to the noise estimate (light gray). Note that a different numeric colorscale is used in each panel.



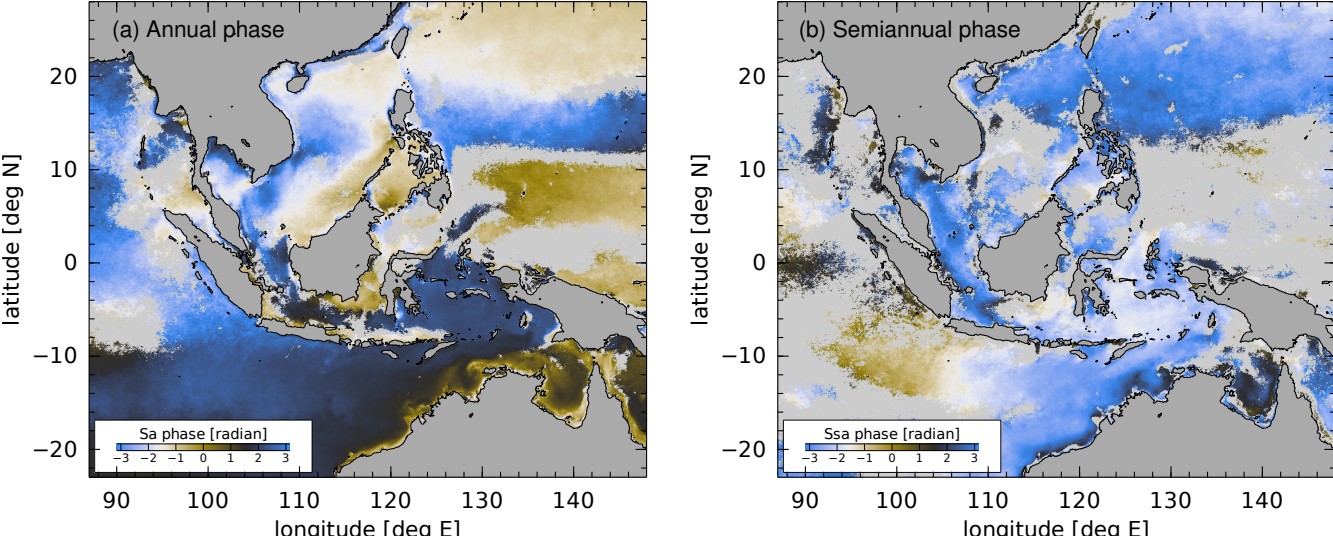

**Figure 4.** The phase of the (a) $S_a$ and (b) $S_{sa}$ harmonic constants. The phase is measured relative to the vernal equinox (March 20). For $S_a$, brown corresponds to a maximum in February-April, black to May-July, blue to August-October, and white to November-January. For $S_{sa}$, there are two maxima per year; brown corresponds March-April and August-September, while black is offset 1.5 months later.

northeast Arafura Sea, and along the southern coast of Java. Although it is of smaller magnitiude, the semi-annual variability is certainly detectable in many of these same regions. In contrast, the significant fortnightly ($MS_f$, Figure 3d) variability of Chl-$a$ is largely isolated to a few distinct sites and regions, including the northwest Australian shelf, the southern part of the Sulu Sea, between the Lesser Sunda Islands, at sites in the Molucca Seas, and in a few other coastal areas. Several of these regions will
be enlarged, below.

To complete the picture of the low-frequency (annual and semiannual) variability, the phases of the $S_a$ and $S_{sa}$ harmonic constants are shown in Figure 4. Throughout the deep ocean, where its amplitude is small, the annual cycle ($S_a$) is characterized by a smoothly-varying phase associated with the winter season poleward of about $20°$ latitude in both hemispheres. Equatorward of $20°$, the annual cycle peaks in June-September (northern hemisphere summer) throughout much of the Indonesian Seas (Su-
santo et al., 2006); however, strong spatial gradients are associated with the northwest Australian shelf and other coastal areas. The seasonal cycle peaks earlier in the year (January-February) near the Island of Borneo, in the Sulu Sea, and further eastward in the Pacific. Although the semiannual variability is smaller, the phase of $S_{sa}$ is, for the most part, stable over large parts of the Indonesian Seas, but it does contain noteable gradients along the Australian coast.

Enlargements of the $MS_f$ amplitude and phase around the Sulu Archipelago are shown in Figure 5 (region indicated in
Figure 1). The largest amplitudes are found in the southern Sulu Sea (on the north side of the archipelago), in a region known for its large-amplitude tidal internal waves (Apel et al., 1985; Zhang et al., 2020; Zhao et al., 2021). Towards the western end of this feature, the amplitude of $MS_f$ is about 1/4 as large as the combined mean, $S_a$, and $S_{sa}$ Chl-$a$, so it represents a significant fortnightly modulation of the Chl-$a$ concentration. The region of maximum $MS_f$ amplitude is associated with a phase lag which



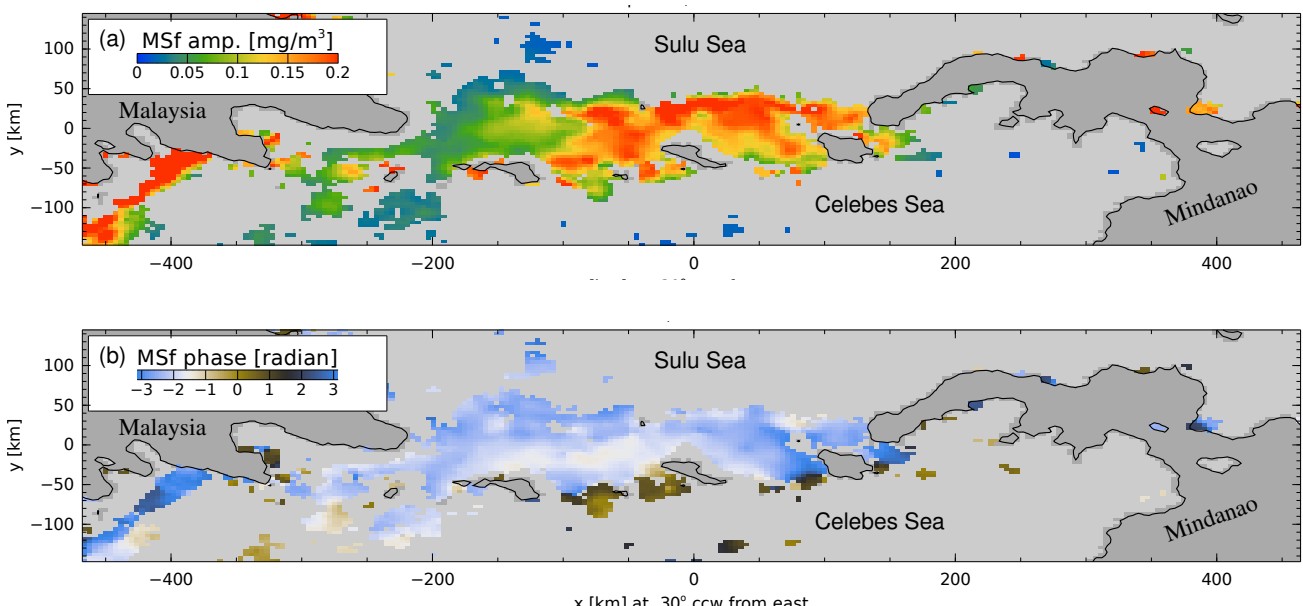

**Figure 5.** (a) Amplitude and (b) phase of the fortnightly MS$_f$ harmonic constants of Chl-$a$ along the southern boundary of the Sulu Sea (indicated in Figure 1).

varies from about $\pi$ radians (7.4 days) in the east ($x = 100$ km) to 4.3 radians (10 days) in the west ($x = -100$ km). The phase

is measured relative to the tidal potential, but Ray and Susanto (2016) show that the barotropic semidiurnal tide at this location lags the potential by about 1.2 days. Thus, the fortnightly maximum in Chl-$a$ occurs from 6.2 to 8.8 days after the maximum spring-neap tides in this region.

Figure 6 presents an enlargement of MS$_f$ Chl-$a$ in the region around the Lesser Sunda Islands, from Bali to Ombai Strait. The maximum in Chl-$a$ occurs between the the islands of Lombock and Sumbawa in the west, and within the northern part of

the Ombai Strait in the east. The phase lag exhibits a relatively strong southward gradient, but close to the islands it is about 3.5 radians (8.2 days) lagged from the astronomical potential. The barotropic tide lags the potential by about 2 days here (Ray and Susanto, 2016), so the fortnightly Chl-$a$ maximum again lags the spring tides by about 6.2 days. The phase propagates southward at a rate of roughly 1.5 radians per 80 km, equivalent to $0.2\mathrm{m/s}$.

## 4   Discussion

The mean and annual variability of Chl-$a$ shown above replicate the results of Susanto et al. (2006), and provide an update with greater resolution based on the longer time series and different satellite missions now available. The use of harmonic analysis to explicitly separate the mean, annual, semi-annual, and fortnightly variability provides an alternative to the climatologies averaged into calendar months or lunar phase used in previous works (Susanto et al., 2006; Shi et al., 2011).





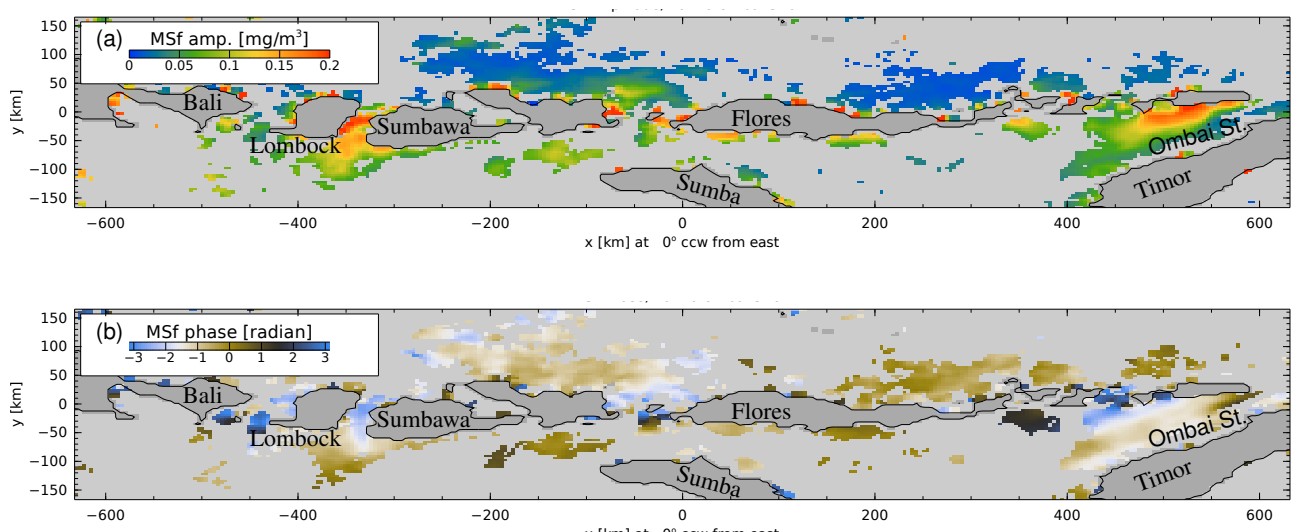

**Figure 6.** (a) Amplitude and (b) phase of the fortnightly $MS_f$ harmonic constants of Chl-$a$ along the Lesser Sunda Islands, from Bali to Ombai Strait (indicated in Figure 1.

The harmonic analysis of Chl-$a$, together with its error estimate, appears to provide more consistent classification of sig-
nificant and non-significant signals, as compared with the wavelet-based approach of (Xing et al., 2021), which indicated significant $MS_f$ variability in relatively large regions of the deep Indian and Pacific Oceans. The $MS_f$ signal in most regions of deep water is not significant (Figure 3d), except for the region southwest of Sumatra which is unexplained. It is interesting that some patches of Chl-$a$ variability, for example, east of Luzon Strait at (20°N,122°E) and above the Mariana Arc system (18°N,146°E), are associated with sites of large baroclinic semi-diurnal tides (Jan et al., 2008; Zhao and D'Asaro, 2011;
Pickering et al., 2015) which could, conceivably, modulate upper ocean nutrient transport with a spring-neap cycle.

The fortnightly maximum of Chl-$a$ lags the spring tide by roughly 6 to 7 days for the examples cited in the Results section, and roughly the same phase lag occurs at other sites which have been checked along the northwest Australian shelf and along the archipelagos in the Molluca Sea. The Chl-$a$ phase lag thus exceeds the 1- to 3-day lag between the spring tide and the SST minimum estimated by Ray and Susanto (2016) in the same southern Sulu Sea and Lesser Sunda Islands sites as considered
here. The potential relationship between Chl-$a$ and nutrient inputs due to tidal mixing is complex (Laws, 2013), but the timing of the fortnightly Chl-$a$ maximum is logically consistent with the co-occurance of vertical nutrient and heat fluxes associated with tidal mixing.

In some respects the spatial distribution of $MS_f$ Chl-$a$ variability coincides with the map of sub-surface tidal dissipation inferred in the numerical model of Nugroho et al. (2018) (their Figure 7), but it is notable that the $MS_f$ Chl-$a$ signal is absent
or confined to very small regions of many sites where the dissipation is large, e.g., in the Strait of Malacca, in the Makassar Strait, in the northern Arafura Sea, etc.



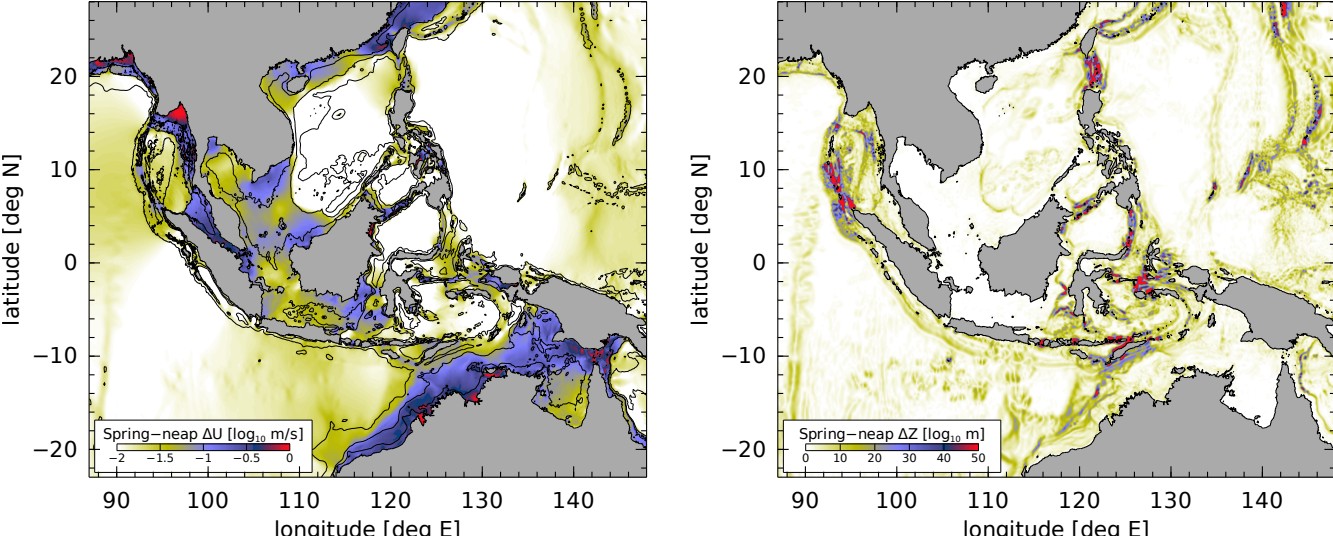

**Figure 7.** Diagnostics of the barotropic tidal currents, based on the TPXO9-Atlas tide model (Egbert and Erofeeva, 2002). (a) The amplitude of oscillation of the combined $M_2$-$S_2$ current at the $MS_f$ period. Solid contours show the coastline, 50 m, 200 m, and 2000 m isobaths. (b) The amplitude of oscillation of the vertical particle displacement due to combined $M_2$-$S_2$ bottom flow across isobaths.

In order to better understand the mechanisms leading to the Chl-$a$ variability, Figure 7 illustrates two diagnostics computed from the barotropic currents predicted by the TPXO9 tidal model (Egbert and Erofeeva, 2002). The first diagnostic is denoted $\Delta U$, and it is the barotropic tidal current speed associated with the spring-neap cycle (Figure 7a). Assuming that the energy source that drives the fortnightly mixing is proportional to the kinetic energy of the combined $M_2$ and $S_2$ tides,

$$KE = \frac{1}{2}(U_1 \cos(\omega_1 t - \phi_1) + U_2 \cos(\omega_2 t - \phi_2))^2, \tag{1}$$

where $U_i$ is the major axis of the tidal current, $\omega_i$ is the frequency, $\phi_i$ is the phase lag, and subscripts 1 and 2 correspond to the $M_2$ and $S_2$ tides, respectively; then the amplitude of the spring-neap cycle of KE is given by the product, $(\Delta U)^2 = U_1 U_2$. In contrast, the baroclinic tide is generated by flow across topography, leading to localized vertical displacements of isopycnals and baroclinic pressure gradients. A metric of this quantity is $\Delta Z = w/\omega_1$, the vertical displacement of a water parcel flowing along the bottom (Figure 7b), where $w = |w_1 w_2|^{1/2}$ is given in terms of the cross-isobath $M_2$ and $S_2$ currents, $w_i = -\mathbf{u}_i \nabla H$, with $\mathbf{u}_i$ the major-axis current vector.

Figure 7a shows that $\Delta U$ is greatly enhanced in several areas where the amplitude of Chl-$a$ $MS_f$ is large, especially along the northwest Australian shelf, and in the Gulf of Martaban at the north end of the Andaman Sea. In contrast, Figure 7b shows that large values of $\Delta Z$ coincide with large-amplitude $MS_f$ Chl-$a$ along the Lesser Sunda Islands and across sills adjacent to the Molluca and Sulu Seas. Just as with the Nugroho et al. (2018) dissipation estimate, it is also notable that there are a number of sites where $\Delta Z$ is large, among the Andaman-Nicobar Islands and within Luzon Strait, where the $MS_f$ Chl-$a$ amplitude is not significant. Indeed many of these same regions are well-known for the presence of nonlinear internal waves (Osborne



and Burch, 1980; Zhao and Alford, 2006), and it is puzzling that a robust $MS_f$ Chl-$a$ signal is not detected at these sites. This
suggests that the fortnightly modulation of Chl-$a$ is not caused solely by the tides, but such a modulation is possible only when
the pre-existing dynamical balance of nutrients and phytoplankton is suitable to control.

## 5  Conclusions

Harmonic analysis of 20 years of daily MODIS-Aqua-derived Chl-$a$ measurements has been conducted to identify regions of
tidal influence on biological production in the Indonesian Seas and nearby Indian and Pacific Oceans. The choice of analysis
frequencies, $S_a$, $S_{sa}$, and $MS_f$, was dictated by the spectral resolution feasible with the time series which suffer from extensive
gaps due to clouds. The results agree with previous estimates of the annual cycle (Susanto et al., 2006), and reveal several sites
and regions where the Chl-$a$ concentration is phase-locked with the spring-neap cycle, peaking around 6-to-7 days after the
local maximum of tidal currents.

The locations of the largest $MS_f$ Chl-$a$ signals, off the northwest coast of Australia and in the vicinity of narrow channels
between islands, generally correspond to regions where boundary-layer-driven and internal wave-driven mixing are expected
to occur, based on comparisons with models. However, there are a number of regions where tidal dissipation is large or where
large-amplitude internal waves are well-known, but where a significant $MS_f$ Chl-$a$ amplitude is not found. This observation
highlights the obvious point that tidal controls on Chl-$a$ occur only at sites where the prerequisites for such control are suitable.

One limitation of the present estimates is the difficulty with identifying the small $MS_f$ Chl-$a$ signal from the gappy data.
Experiments to improve the estimates were conducted using spatially-coupled harmonic analysis (not shown); however, results
were found to be overly sensitive to details of the spatial basis functions used. Ocean models with the capability of simulating
the tidal-biological interactions are now being implemented (e.g., Gutknecht et al. 2016 and Capuano et al. 2022), and it may
be fruitful to use the surface Chl-$a$ fields from these models as spatial basis functions within a spatially-coupled harmonic
analysis, as was recently done for sea surface-height data (Egbert and Erofeeva, 2021). While imperfect, such an approach may
provide the capability to utilize the gappy data more effectively to map tidally-phase-locked components of the Chl-$a$ fields.

*Code and data availability.*   The Level-3 GlobColour product may be obtained from https://oceancolor.gsfc.nasa.gov/. Julia-language scripts
for harmonic analysis of these data are available from the software repository at https://ingria.ceoas.oregonstate.edu/fossil/CHL.

*Author contributions.*   Zaron drafted the manuscript, and implemented and performed the harmonic analysis. Capuano provided the Chl-$a$
dataset and contributed to revising the manuscript. Koch-Larrouy conceived of the research, guided the selection of significant geographic
regions for study, and contributed to revising the manuscript.

*Competing interests.*   The authors have no competing interests in relation to this work.



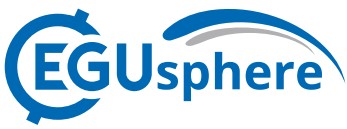

*Acknowledgements.* Support for Zaron was provided by the NASA Ocean Surface Topography Science Team, award #80NSSC21K1189.



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
