# Peer review of "Tidal Variability of Chl-$a$ in the Indonesian Seas"

_EGUsphere, 2022_

## Author Comment (AC1)

**Reply to Reviewers, re: egusphere-2022-821**

We appreciate the time taken by the reviewers to read the manuscript and provide comments and questions, and we especially appreciate that their comments were posted so promptly after submission. The reviewers' comments are reproduced below in black, and our replies to their queries are typeset in *blue*.

Note that the title of the article has been changed from "Tidal Variability of Chl-*a* in the Indonesian Seas" to "Fortnightly Variability of Chl-*a* in the Indonesian Seas", in order to more precisely characterize the results.

**Reviewer #1:**

1. The authors try to convince the reader that harmonic analysis of Chl-a over the Indonesian seas may show spring-neap variability associated with tidal currents and baroclinic tides, which may provide evidence of strong tidal mixing in the region. They also show the semiannual and annual cycles. The logical structure of the presentation and the writing are very good; however, in my opinion, at this current stage, it is not ready for publication at the AGUSphere. They focus more on data gaps and processing and lack physical and dynamic mechanisms that can motivate the reader to understand the relationship between tidal harmonic and chlorophyll-a concentration and tidal mixing. There is only one line (line 125) in Section Results, "the potential relationship ...". This should be more elaborate in the Introduction and Discussion sections.

*These are fair remarks to the contents of the article, and we have conducted two new analyses in response, adding: (1) an expanded discussion of methodological issues to an appendix, focussing on the role of data gaps in the harmonic analysis of surface Chl-a, and (2) evidence of nonlinear baroclinic overtides at the $MS_4$ frequency from satellite altimetry. The latter contributes to the dynamical picture of potential mechanisms to explain the spring-neap cycle of Chl-a, since it is a direct measure of nonlinear interaction between astronomically forced tides at the $M_2$ and $S_2$ frequencies.*

*Specific changes to the manuscript regarding the Chl-a dynamics include the following: (1) The Introduction has been expanded to discuss the general context for investigating this region, especially the influence of tidal mixing and monsoons on physical processes in the Indonesian Seas; (2) an additional sentence was placed at the end of the fourth paragraph (line 45) to acknowledge that this paper is indeed focussed on describing the observational evidence for the effects of tide/Chl-a interaction, and we do not have direct evidence regarding the mechanisms, which can only be obtained from studies in situ; and (3) a new section was added in the Discussion which shows evidence for nonlinear overtides in satellite altimetry, produced by the nonlinear interaction of $M_2$ and $S_2$ tides.*

2. Southeast Asia Seas/Indonesian seas are strongly influenced by monsoons which drive seasonal variability of the ocean dynamics and climate, seen in the SST, chlorophyll, rainfall, etc. (see, i.e., Aldrian and Susanto, 2003). They present semiannual and annual Chl-a cycles. However, in the paper, no mention of or word on monsoon. Had they collaborated with local scientists may help in the results/interpretation of oceanic and

[Figure]

Figure 1: Amplitude of $MS_f$ Chl-*a* along the Lesser Sunda Islands, from Bali to Ombai Strait (cf., Figure 5 in the manuscript), based on data from (a) the wet northwest monsoon (year-days 240 to 95), and (b) the dry southeast monsoon (year-days 85 to 250)

atmospheric conditions of Indonesian seas. During the northwest monsoon (boreal winter; wet season), most regions will be covered by clouds and much more clouds/rainfall during La Nina. Hence, during the northwest monsoon from October to April (6 months), it is hardly seen a reliable daily map of chl-a. Hence, Figure 1 and their results may skew toward the southeast monsoon (dry season). Maybe they may have to divide the data availability based on monsoon seasons. Ray and Susanto (2019) show that atmospheric tides may be due to ocean tide (air-sea interaction) and vary with the monsoon. Susanto and Ray (2022) recently showed that tidal mixing in the Indonesian seas varies with the monsoon, ENSO and IOD.

*The reviewer points out the lack of discussion of monsoon processes in the results, which is especially relevant to the interpretation of annual and semi-annual variability. The correlation of the monsoon with the seasonal cycles and cloudiness also suggests that data gaps could bias the harmonic constants. We agree that these are important points. We intentionally limited our initial analysis to a superficial examination of the annual cycle, mostly to replicate and update the previous results of Susanto et al (2006), since a discussion of monsoonal influence on Chl-a is beyond our scope.*

*We have made two significant modifications in response to the reviewer's comments regarding the monsoons: (1) as already mentioned, the Introduction has been expanded to provide more context for the role of the Indonesian Seas in the climate system, and acknowledge the significance of monsoonal processes, (2) we have added an appendix to examine the significance of missing data which follow the monsoon cycle.*

*In addition, we have computed $MS_f$ harmonic constants separately for the dry southeast monsoon and wet northwest monsoon seasons. The separate analyses are obviously based on smaller datasets, and exhibit more grid cells of marginal significance offshore (Figure 1, top of this page). One potentially significant difference between the monsoonal seasons is the strength of the Chl-a band along the southern coast of Flores.*

*As mentioned in the reply to the next query, we think an appropriate way to study the monsoonal modulations of tides and Chl-a would follow the approach in Mandal et al 2022, which generalizes the notion of harmonic analysis (regression onto sines and cosines of a given frequency) to a regression approach that explicitly incorporates monsoonal variability.*

3. Does their seasonal harmonic analysis show the seasonal Chl-a due to tidal frequency at seasonal cycles, monsoon, or both? If both, can we separate them?

*These are certainly good questions. The annual and seasonal cycles are simply defined by their periodicities relative to the solar year. In contrast, the monsoons are defined as recurring phenomena with periodic and quasi-periodic expressions in multiple physical domains (winds, precipitation, etc.), driven by land-sea temperature contrasts.*

*The question of whether the seasonal and monsoonal cycles can be separated is best addressed using multi-component data, since the seasonal and monsoonal phenomena are categorically different concepts. Analyses of monsoonal variability using regression techniques and non-periodic basis functions, such as Mandal et al (2022; see reviewer's suggested references, below), seem to be the most appropriate way to address these questions. This topic is beyond the scope of this manuscript, though, but we now cite the work of Mandal et al (2022) as an exemplar of such an approach.*

*An extension of the reviewer's questions leads naturally to the methodological issue of whether missing data, which is correlated to monsoonal processes, could significantly corrupt or bias the harmonic constants. We have investigated this question in the newly-added Appendix by performing a harmonic analysis of a gap-free daily time series containing significant monsoonal variability associated with the annual and semi-annual periodicities, namely, the NCEP reanalysis winds at the 850mb level at $(7.5°S,110°E)$. Then, the harmonic analysis was repeated for this given time series, but omitting data from the same days as the Chl-a time series. This allowed us to examine the effect of the missing data with a known time series, and showed that the semi-annual cycle cannot be accurately determined here.*

4. I am curious about high MSf amplitude in the northern coasts of the Lesser Sunda Island (LSI) and what mechanism generates these features. Is it due to monsoon or other dynamical processes? During the southeast monsoon, upwelling (high chl-a concentration) occurs along the southern coasts of the LSI. Meanwhile, upwelling during the northwest monsoon occurs along the LSI's northern coasts, and downwelling during the southeast monsoon (i.e., Wirasatriya et al., 2021). Similarly, seasonal upwelling occurs in the Malacca Strait (i.e., Mandal et al., 2021).

*We appreciate the reviewer's detailed queries about the region north of LSI. We now believe that these small signals were an artifact of the data gaps. As explained in the new Appendix, the data gaps can alias broadband (non-periodic) low-frequency variability into the $MS_f$ frequency. The variability north LSI is now regarded as non-significant and grayed out in the new figures.*

*We do note from Figure 1, above, that the $MS_f$ variability along the southern coastline of the LSI is enhanced during the southeast monsoon. As implied by the reviewer's comments, this observation could not really be explained without considering the monsoons, so we have not discussed this in the manuscript.*

5. They discuss the amplitude and phase of the Chl-a spring - neap tide. Please add

a more physical and dynamic mechanism that relates the tide and the peak of Chl-a.

*A new analysis of satellite altimetry data has been added to the Discussion which clearly shows the presence of baroclinic overtide waves at the MS$_4$ frequency. These waves are strong evidence of nonlinear interactions of the astronomically-forced M$_2$ and S$_2$ waves, which adds to the already extensive evidence from the INSTANT and INDOMIX programs, numerical models, and SAR backscatter. The fortnightly variability in Chl-a is the low-frequency counterpart of the high-frequency variability.*

*Determining the physical mechanism which relates Chl-a and tides, among the possibilities mentioned at line 39 where Xing et al (2021) is cited, is beyond the scope of this paper. Our point here is to document the fortnightly variability of upper ocean Chl-a in order to identify specific locations where further research could be conducted or where comparison with numerical models might be useful.*

Minor:

Ref. Capuano et al., 2022 cannot be accessed because it is in the submission process. I am not sure about the rule of EGUsphere.

*We have have removed the reference to Capuano et al 2022, since it has not yet been accepted for publication.*

Link to the software repository does not work.

*We apologize for this oversight. The repository is now populated and working.*

Some references below may be relevant to the topic which may be added in the citation:

*Thank you very much for these pointers to the literature, especially the most recent papers.*

Susanto, R. Dwi, and Richard D. Ray, Seasonal and interannual variability of tidal mixing signatures in Indonesian seas from high-resolution sea surface temperature, Remote Sensing, 2022, 14, https://doi.org/10.3390/rs14081934

*This reference is now cited in the Results regarding differences in Chl-a MS$_f$ between the dry southeast monsoon and wet northwest monsoon.*

Mandal, Samiran, Susanto, R. Dwi, and Balaji Ramakrishnan, Dynamical Factors Modulating Surface Chlorophyll-a Variability along South Java Coast, Remote Sensing, 2022, 14, 1745. https://doi.org/10.3390/rs14071745.

*This reference is now cited in the Introduction, regarding the separation of seasonal and monsoonal variabilities via non-harmonic analysis, i.e., regression onto non-periodic basis functions such as climate indicators.*

Mandal, S., N. Behera, P. C. Pandey, A. Gangopadhyay, and R. Dwi Susanto, Evidence of a Chlorophyll "Tongue" in the Malacca Strait from Satellite Observations, J. Marine Research, 2021, 233, November, https://doi.org/10.1016/j.jmarsys.2021.103610

*This paper is now cited in the discussion of the mean and seasonal Chl-a maps in the Results.*

Wirasatriya A., R. Dwi Susanto, Kunarso, A. R. Jalil, F. Ramdani, A. D. Puryajati, 2021. Northwest Monsoon Upwelling Within the Indonesian Seas. International Journal of Remote Sensing, 42:14, 5437-5458, DOI: 10.1080/01431161.2021.1918790

*This paper is now cited in the Introduction regarding the profound influence of monsoon winds on ocean processes throughout the Indonesian Seas.*

Siswanto Eko, Takanori Horii, Iskhaq Iskandar, Jonson Lumban Gaol, Riza Yuliratno Setiawan, R. Dwi Susanto, Impacts of climate changes on the phytoplankton biomass of the Indonesian Maritime Continent, Journal of Marine Systems, 2020,103451, ISSN 0924-7963, https://doi.org/10.1016/j.jmarsys.2020.103451.

Ray, R. and R. D. Susanto, 2019: A fortnightly atmospheric 'tide' at Bali caused by oceanic tidal mixing in Lombok Strait, Geoscience Research Letter, 6:6, https://doi.org/10.1186/s40562-019-0135-1

*This reference is now cited in the Appendix, in the discussion of the power spectrum of the data indicator function.*

Aldrian, E. and R. D. Susanto, 2003: Identification of three dominant rainfall regions within Indonesia and their relationship to sea surface temperature, International Journal of Climatology, 23, 12, 1435-1452, doi: 10.1002/joc.950.

*This reference is now cited in the Appendix in relation to the definition of the monsoon seasons.*

**Reviewer #2:**

The authors performed a harmonic analysis of 20 years of MODIS chl data in the Indonesian Seas and concluded that "in some regions chl is phase locked with the spring-neap cycle, peaking around 6-7 days after the local maximum of tidal currents". It is an interesteing topic, but in my assessment the paper needs much more work before it is publishable.

As the authors clearly state, the time series of chl only has data at most 50% of the time due to cloud coverage, and in many regions it is as low as 7-14%. I am not convinced that this is sufficient temporal coverage to support the finding of a fortnightly signal. The authors state this it is sufficient but present little actual analysis to back this up. Given this is a region strongly impacted by monsoons (which is not mentioned in the paper) I imagine that there could be quite long periods with no data. They need to do a better job of assessing the gaps in the data and convincing the reader that it is not problematic to answer the question posed here.

*The reviewer makes valid points, and we have added an Appendix which delves into the issue of missing data more thoroughly. This analysis was particularly useful for helping us to clarify the distinct roles played by the data gaps in comparison to the broadband variability not represented by the harmonic decomposition. The broadband variability (which includes both measurement error and non-periodic signals) contributes to the estimation error of the harmonic constants, and has been the subject of diverse approaches to estimation and significance testing in the harmonic analysis literature (Pawlowicz et al 2002, Foreman et al, 2009). The role of data gaps has been less studied in the literature, and, to our knowledge, the analysis of estimation error in the appendix is a new result; although, it is closely related to classical results from linear estimation.*

*We agree that the monsoon-related processes – such as coastal upwelling, wind-driven mixing, precipitation-induced stratification, runoff and river-plumes, and cloud-penetrating solar insolation – are likely to have significant influences on upper ocean Chl-a. We have added three paragraphs to the Introduction to better contextualize our*

*work and acknowledge the importance of monsoon processes.*

*While the monsoons are of over-riding significance throughout the Indonesian and neighboring Seas, there is an already vast literature on the subject. Monsoon processes are particularly complex due to their geographic variability and timing, which we did not wish to address in this manuscript. We have maintained the article's focus on the relatively high-frequency fortnightly variability of Chl-a.*

I was curious why they used the GlobColour MODIS product rather than their merged product. The GlobColour project merges satellite data from different sensors which should result in a product with slightly better spatial coverage. Additionally, there are known issues with recent data from the MODIS sensor, which is 15 years past its design life. Using a merged product like GlobColour or the ESA OC-CCI product would mitigate the impact of the declining data quality of the MODIS data.

*We approaciate the reviewer's suggestion, but we actually tested a few products available for Chl-a sea surface data (the MODIS GlobColour daily product and their merged version; and also the CMEMS_CCI product) and found that the most appropriate one for our study was the first one. Indeed, the merged products, despite the fact of presenting a slightly improved data coverage, show a smoothed signal and in the case of the harmonic analysis we carried out, we were not able to capture the tidal variability signature on the Chl-a field when using such interpolated products. Moreover, we did not perceive any problem in terms of data quality when assessing the performance of the MODIS daily product, in comparison to the merged one, in our region of interest, particularly within the Indonesian straits and island chains which are hotspots of strong internal tide mixing.*

They need to do a better job of introducing the study area and describing what is already known about tidal variability in this area, and what big questions still need to be answered. They don't even mention the study area in the abstract except for in the last sentence. Reading just the abstract one might assume this was a global analysis that happened to highlight dynamics in the Indonesian Seas. Additionally, the conclusion cited above came from the Conclusion section, the abstract was much more vague on the actual results of the paper.

*We have extensively revised the abstract and introduction in response to both reviewer's comments.*

There are a lot of details about the data analysis or their results that are glossed over or are vague:

Lines 34-35. "We have found that least-squares harmonic analysis is adequate to quantify variability"

How was this assessed? What is their definition of "adequate"? What would be inadequate?

*We agree that this is a vague statement, but it is one of the last sentences of the Introduction and it is made precise by the subsequent analysis in the manuscript. It is placed at the end of a series of sentences which discuss other methodological approaches from the literature, and contrast with our approach, harmonic analysis, which we believe to be more simple and direct than the previous ones.*

Lines 65-67. They state that the noise of the harmonic constants was compared to the amplitude at "fake" tidal frequencies and found to agree well.

[Figure]

Figure 2: Comparison of harmonic analysis errors estimated from amplitudes at "fake", non-tidal, frequencies (y-axis) versus the standard error estimated from the pseudo-spectrum of the residual (x-axis). The linear fit (red) indicates that the two approaches agree with an offset of about 15% over three orders of magnitude range in error.

What is meant by the "amplitude estimated at 'fake' tidal frequencies"? Compared how? What is their definition of "agree well"?

*Clarifying text has been added at line 85 to state the exact values of the fake frequencies, "The fake frequencies used were $S_a$ plus 2 cycles-per-18-years (Doodson number 0 567 555) and MSf minus 3 cycles-per-year (Doodson number 0 205 555)." A number of different frequencies were tested, and these were simply the last in the series of experiments.*

*The amplitude of the fake $S_a$ harmonic constant is compared with its standard error estimate in Figure 2, above. One can see that the two quantities are linearly related over 3 orders of magnitude. The root-mean-square of the fake amplitude is about 15% smaller than the standard error estimate. This offset (which applies to the error estimate, not the harmonic constant, itself) is sensitive to the shape of the residual spectrum near the analyzed frequencies, so it varies with the band-averaging used to compute the pseudo-spectrum in the standard error calculation, and the choice of fake frequencies, but it seems to be accurate enough for a useful error estimate.*

Lines 79-80 "it is evident that the seasonal variation of chl-a is 50% or more of the mean in many coastal areas". I'm afraid this is not evident to me. Is this from a comparison of the mapped values in Fig3a and 3b? The two panels have different color scales which makes comparisons between them more difficult.

*The range shown in the original Fig 3a was approximately twice as wide as the range shown in Fig 3b. As the colorscales are both saturated, or nearly so, in many coastal areas, the amplitude of $S_a$ (Fig 3b) is roughly 50% of $Z_0$ (Fig 3a). No change to text. Note that the colorscales have been changed slightly to improve the legibility in this revision,*

*and the colorscale in Fig 3a is exactly twice the range of the colorscale in Fig 3b.*

Lines 114-115 "The harmonic analysis... appears to provide more consistent classification..."

Based on what? What is meant by "consistent classification"?

*The discussion at this point in the manuscript is regarding the fortnightly cycle estimated by Xing et al (2021) using wavelets, compared to the harmonic analysis. Their analysis found fortnightly variability in rather large portions of the deep ocean for which there is no plausible physical explanation. The approach to sigificance testing used here, based on the pseudo-spectrum of the residual time series, flags these regions as non-significant, so we believe the results here are more reliable and provide a "consistent classification" of significant and non-significant signals.*

*The phrase "consistent classification" has been changed to "reliable classification".*

They frequently make comparisons to their results and those of others that require some degree of graphical representation, i.e. lines 117-119, 128-129, 114-115.

*We agree that the comparisons with previous works does indeed benefit from a visual or graphical comparison to efficiently identify the differences among the results. We merely wish to point out some of the differences we identified, and highlight why we think our results are better. The reader interested in such details will have to examine these differences graphically or quantitatively, depending on their level of interest.*

Line 121. "The fortnightly Chl maximum lags the spring tide... for the examples cited in the Results Section"

They need to be more specific about what results (or areas) they are referring to.

*This sentence was revised to, "The fortnightly maximum of Chl-$a$ lags the spring tide by roughly 6 to 7 days for the examples cited previously in the Southern Sulu Sea and along the Lesser Sunda Islands, ..."*

Lines 157-158. "in some regions chl is phase locked with the spring-neap cycle, peaking around 6-7 days after the local maximum of tidal currents". Where is this shown? What figure?

*This was shown in Figures 4 and 5 and discussed in the paragraphs from lines 126 to 140 in the revised manuscript. Note that the day range, "6-7", has been changed to "6-9" to match the text at line 133.*

Figure 1. Convert the values shown to percentages.

*Done.*

Fig 4, 5 and 6. It would be easier for readers to interpret the maps if the colorbar legends for phase were expressed as months rather than radians.

*Agreed. The phase is now represented as months for $S_a$ and days for $MS_f$.*

*Note that we have corrected a typo in the units of the Fig 6 colorbar (removed "$\log_{10}$").*

---

## Author Response (AR2)

**Reply to Editor and Reviewers, re: revision 1 of egusphere-2022-821**

Once again, we appreciate the time and interest of the reviewers in carefully reading this manuscript. The reviewers' comments are reproduced below in black, and our replies to their queries are typeset in *blue*.

**Article Validation, Polina Shvedko**

Notification to the authors:

The title page of pdf manuscript file must include the full institutional addresses of all authors. However, the address for affiliation #2 looks like the it is not completed. Please enter full information for the next revision.

*Done.*

**Reviewer #1:**

The authors have answered and revised my questions and concerns. Therefore, I do not have any major suggestions and comments except two minor comments below that should be clarified before ready for final publication.

1. In the paragraph 175 (Discussion). Could you provide some clues and speculate for the possible reasons why some regions with large delta-Z (i.e. along the coasts of China) but no significant Chl-a concentration, while the tidal mixing from SST (i.e., Susanto et al., 2019) and numerical model (Nugroho et al., 2016) showed strong tidal mixing signatures.

Susanto, R.D.; Pan, J.; Devlin, A.T. Tidal Mixing Signatures in the Hong Kong Coastal Waters from Satellite-Derived Sea Surface Temperature. Remote Sens. 2019, 11, 5.[1]

*The analysis of Susanto et al 2019 is concerned with SST signatures in the region around Hong Kong and the Pearl River Estuary. They use spectral analysis to analyze SST data and note variance at the spring-neap periodicities associated with M2-S2 and K1-O1 tides. Their SST analysis is bolstered by in situ data, which reveals the same modulations of currents and vertical shear.*

*The reviewer's comment states that some regions along the coast of China have large delta-Z, but this is mistaken and not indicated by Figure 6b. Perhaps this was a typo in the reviewer's comment. In any case, it is indeed interesting that tidal modulations of SST in that area are evidenced in Susanto et al 2019 and Nugroho et al 2016, but not in our work on Chl-a.*

*We speculate that an explanation for this discrepancy could be the low signal-to-noise of tidal Chl-a modulations along this part of the Chinese coast. Based on the power spectra shown in Susanto et al 2019, the SST peaks at Mf and MSf periods barely exceed the background level (their Figure 5); although, they did pass the significance test for mapping the harmonic analysis which ought to be very similar to the test we used (their Figure 4).*
* * *
[1]https://doi-org.proxyum.researchport.umd.edu/10.3390/rs11010005

*Overall, investigation of Chl-a in coastal areas such as the Pearl River Estuary is probably complicated by anthropogenic influences, especially the eutrophication of coastal waters (e.g., see discussion and references in Zhang et al, 2022, "Categorizing numeric nutrients criteria and implications for water quality assessment in the Pearl River Estuary, China", Front. Mar. Sci.[2] The average nutrient concentration of these waters may be such that the tidal modulations of mixing do not result in modulations of Chl-a, even if they are visible in physical parameters such as SST.*

2. Susanto, R. Dwi, and Richard D. Ray, Seasonal and interannual variability of tidal mixing signatures in Indonesian seas from high-resolution sea surface temperature, Remote Sensing, 2022, 14.[3]

Author response: This reference is now cited in the Results regarding differences in Chl-a MSf between the dry southeast monsoon and wet northwest monsoon.

» Please recheck, I do not see any of this reference in the Results nor entire manuscripts.

*We apologize for the oversight; the reference was added and deleted from intermediate drafts. We feel that it is important to direct readers to that work, because of its discussion of monsoonal modulations of the spring-neap cycle of SST. The reference has now been added to the end of the sentence at line 31.*

**Reviewer #2:**

This manuscript is much improved! They did a great job revising the abstract and the introduction to set the reader up for what the paper will be presenting. I have a few minor points, listed below, that would improve the manuscript. Overall I feel it is ready for publication now.

*Thank you. Your comments and those of the other reviewer were significant and appreciated. They are now noted in the Acknowledgements.*

Line 41/42; "could explain the association of remotely-sensed Chl-a with spring-neap currents" rewrite as "could explain the association of increased Chl-a with spring-neap currents". It's the increase in chl that is significant, not that it is measured by satellites.

*Agreed.*

Or is it changes in chl? The second possible mechanism listed, the resuspension of sediments reducing PAR, would reduce chl. Clarify what is meant here.

*The parenthetical phrase, "(which ought to decrease apparent Chl-a)", has been added to clarify.*

Figure 1. Consider adding a point to indicate the area of the data analysis shown in Figure 2.

*Done.*

[2]https://doi.org/10.3389/fmars.2022.1004235)
[3]https://doi.org/10.3390/rs14081934